# HDL in Atherosclerotic Cardiovascular Disease: In Search of a Role

**DOI:** 10.3390/cells10081869

**Published:** 2021-07-23

**Authors:** Manuela Casula, Ornella Colpani, Sining Xie, Alberico L. Catapano, Andrea Baragetti

**Affiliations:** 1Epidemiology and Preventive Pharmacology Service (SEFAP), Department of Pharmacological and Biomolecular Sciences, University of Milan, 20133 Milan, Italy; manuela.casula@unimi.it (M.C.); ornella.colpani@unimi.it (O.C.); sining.xie@unimi.it (S.X.); 2IRCCS MultiMedica, Sesto S. Giovanni, 20099 Milan, Italy; alberico.catapano@unimi.it; 3Department of Pharmacological and Biomolecular Sciences, University of Milan, Via G. Balzaretti, 9, 20133 Milan, Italy

**Keywords:** HDL cholesterol, HDL lipoproteins, genetics, pharmacological trials

## Abstract

For a long time, high-density lipoprotein cholesterol (HDL-C) has been regarded as a cardiovascular disease (CVD) protective factor. Recently, several epidemiological studies, while confirming low plasma levels of HDL-C as an established predictive biomarker for atherosclerotic CVD, indicated that not only people at the lowest levels but also those with high HDL-C levels are at increased risk of cardiovascular (CV) mortality. This “U-shaped” association has further fueled the discussion on the pathophysiological role of HDL in CVD. In fact, genetic studies, Mendelian randomization approaches, and clinical trials have challenged the notion of HDL-C levels being causally linked to CVD protection, independent of the cholesterol content in low-density lipoproteins (LDL-C). These findings have prompted a reconsideration of the biological functions of HDL that can be summarized with the word “HDL functionality”, a term that embraces the many reported biological activities beyond the so-called reverse cholesterol transport, to explain this lack of correlation between HDL levels and CVD. All these aspects are summarized and critically discussed in this review, in an attempt to provide a background scenario for the “HDL story”, a lipoprotein still in search of a role.

## 1. Introduction

Current knowledge on the relevance of high-density lipoproteins and their cholesterol content (HDL-C) in the initiation and evolution of atherosclerotic cardiovascular diseases (CVDs) has dramatically changed over recent years.

In epidemiological studies, low levels of HDL-C are predictive of an increased risk of CVD, and this has led to searching for a role for HDL in modulating pathways involved in atherosclerosis. For several years, the most popular suggested role has been the so-called reverse cholesterol transport, a metabolic pathway by which HDL can remove cholesterol form lipid-laden cells present in the arterial wall and deliver it to the liver for excretion, at least in part, with the bile. These findings have prompted efforts to discover pharmacological therapies aimed at increasing HDL-C levels. The outcomes of trials using drugs increasing HDL-C by different mechanisms, however, have been somehow disappointing and raised the question of whether simply increasing HDL-C may represent the appropriate approach. More recent evidence documenting the heterogeneity of HDL subfractions in terms of size, structure, apolipoprotein content, and function(s) in the atherosclerotic process supports the concept that specific subfractions endowed with distinct properties may be crucial in determining the role of HDL, and that their abundance is not necessarily reflected by the HDL-C level, and therefore modulation of HDL levels may not be positive per se if the (sub)populations of HDL endowed with the specific protective activity are not affected.

In this review, we briefly summarize recent data from observational, epidemiological, genetic, and Mendelian randomization studies linking HDL to CVD; we also analyze the outcomes from trials with HDL-C-increasing strategies (e.g., CETP inhibitors) and trials with reconstituted HDL or apolipoprotein A-I to better understand the role of the “HDL story” in CVD, and to propose future pathways of research.

## 2. The Role of HDL in Lipid Metabolism

HDL are heterogeneous by size, lipid, and protein content [1]. These differences allow discriminating, on the basis of ultracentrifugation, two main subfractions, HDL2 and HDL3, which can be further subfractionated into five subpopulations (HDL2b, HDL2a, HDL3a, HDL3b, and HDL3c). On the basis of electrophoretic mobility or nuclear magnetic resonance (NMR), a different subset of particles can also be identified such as pre-β1/β2/β3 particles and α1/α2/α3/ particles, which not only differ in size but also in shape. Additionally, NMR-based lipidomic analysis of cholesterol and fatty acid contents [2], and proteomic characterizations of apolipoproteins in HDL (differentiating particles containing only apolipoprotein A-I (apoA-I, LpA-I) or both apoA-I and apoA-II (LpA-I/A-II) [3]) are useful approaches to provide insights into the molecular features and the pathophysiological profile of HDL subfractions (Figure 1).

This heterogeneity, although it may be influenced by the methods used for HDL isolation (especially those dependent on salt gradients such as ultracentrifugation), is mainly dependent upon the extensive remodeling that HDL (as with other lipoproteins) undergo in the circulation, due to the activity of several peripheral/hepatic lipases, circulating enzymes (lecithin-cholesterol acyltransferase (LCAT)), and proteins mediating the exchange of lipids, such as cholesteryl ester transfer protein (CETP). After being synthesized in the liver and intestine, apoA-I (the main HDL protein) is secreted in a poorly lipidated form and interacts with cells, acquiring phospholipids (PL) and free cholesterol (FC), thus generating discoidal pre-β-HDL. These “nascent” particles then increase their lipid content through the reverse cholesterol transport process, in which nascent HDL uptake free cholesterol and phospholipids from cells and peripheral tissues, thanks to multiple mechanisms and receptors differentially expressed over tissues. The ATP-binding cassette transporters A1/G1 (ABCA1/ABCG1), expressed on the cell membranes of peripheral tissues, play a major role in this process, via a concentration gradient- and energy-dependent mechanism. HDL also interact with the scavenger receptor class B type 1 (SR-BI), a horse hoof-like transmembrane glycoprotein that binds to HDL and forms a hydrophobic channel for the trafficking of cholesterol; this transport/exchange is gradient-driven, meaning HDL can either download or upload cholesterol from cells. Loss-of-function mutations in the gene encoding SR-BI (SCARB1) cause an increase in HDL-C [4,5] but do not reduce the risk of coronary artery disease [4]. Nascent HDL are believed to acquire cholesterol also by interacting with other transporters, including ABCG5/ABCG8, complex heterodimers responsible for the biliary and trans-intestinal secretion of cholesterol and dietary sterols.

As LCAT esterifies FC within HDL, the nascent discoidal pre-β-HDL become quasi-spherical, giving rise to α-HDL, the major form of circulating HDL. In addition to LCAT, phospholipid transfer protein (PLTP) and CETP contribute to HDL remodeling, by the removal of cholesteryl ester (CE) and glycerolipid contents, in exchange for TG with triglyceride-rich lipoproteins (TGRLs). As HDL3 subfractions are enriched with these lipids, they become larger, spherical HDL2 subfractions, which are then remodeled by peripheral and hepatic lipases [6].

The heterogeneity of HDL not only reflects the complex interaction with different molecular players controlling cellular and extracellular cholesterol but also identifies their participation in multiple pathophysiological functions. By accepting cholesterol from cholesterol-enriched cells, thus modulating the cell membrane and possibly also the intracellular pool of cholesterol, HDL play a major role in the reverse cholesterol transport but also promote antioxidant and anti-inflammatory activities [1,7] (e.g., at the glomerular and tubular level [8,9]), display endothelial/vasodilatory, antithrombotic, and cytoprotective functions, and participate in modulating immune responses to pathogens [10]. All these effects are likely to be linked to the capacity of HDL to modulate membrane cholesterol and, therefore, the aggregation of receptors/membrane proteins in lipid rafts and coated pits, a key process in the activation/modulation of a number of cellular events; furthermore, HDL are involved in glucose homeostasis [11,12]. All together, these activities support the hypothesis that the lipid/apolipoprotein content or HDL size might be targeted to favor the cellular, metabolic, and anti-inflammatory activities of HDL. How these effects can be applied to cardiovascular prevention is less clear because recent epidemiological studies, genetic approaches, and pharmacological trials are not consistent with the notion of the high HDL-C/low CV risk.

## 3. Association between HDL-C and Cardiovascular Risk: Epidemiology

The common notion that HDL-C is inversely associated with the risk of cardiovascular events stems from several epidemiological studies [13,14,15]; in observational studies, each 1 mg/dL (0.026 mmol/L) increase in HDL-C was reported to associate with a 2% decreased risk of coronary artery disease in men and a 3% decreased risk in women, independently of age, blood pressure, smoking, body mass index, and low-density lipoprotein cholesterol (LDL-C). Low HDL-C levels (25 mg/dL) were associated with an increased CV risk, whereas high HDL-C levels (≥65 mg/dL) conferred cardiovascular protection, and these associations were confirmed both at low (100 mg/dL) and high LDL-C concentrations (220 mg/dL) [16], although the association was stronger with higher levels of LDL-C. This observation supported the hypothesis of the cardiovascular protective effects of HDL and prompted HDL-C measurement for CVD prediction [17]. In many of these studies, the highest levels reported were in the range of 80–90 mg/dL. When exploring higher values, as in recent large-scale prospective cohorts, a “U-shaped” curve was reported, with an increased CVD risk at high HDL values [18,19,20]. Although some evidence supporting this hypothesis may have already been derived from previous epidemiological studies [14], it is possible that the wide variability of estimates of the CV risk associated with elevated HDL-C values has held authors back from highlighting this finding.

An analysis of the Multi-Ethnic Study of Atherosclerosis (MESA) cohort provided one of the first signals that very high HDL-C may be correlated with adverse cardiovascular outcomes [21]; in this cohort, a significant increase in the risk of major atherosclerotic CV events was observed for HDL-C levels >80 mg/dL. Then, studies from larger cohorts gathered comparable data to this first single-center experience. The Cardiovascular Health in Ambulatory Care Research Team (CANHEART) dataset [22] showed a statistically significant increased risk of all-cause mortality both with HDL-C levels >80 mg/dL and with HDL-C levels <30 mg/dL compared to a reference range of HDL-C levels of 40–50 mg/dL. These data were gender-wise, since the same risk in women was estimated by even higher HDL-C threshold values (>90 mg/dL).

The Danish Copenhagen City Heart Study and the Copenhagen General Population Study provided the most robust level of evidence [19]. In these two cohorts, comprising over 50,000 men and 60,000 women followed for up to six years, HDL-C levels, assessed as a continuous variable using restricted spline curves, demonstrated a significant “U-shaped” association with all-cause mortality in both men and women, driven by extremely low and high concentrations. From then on, multiple studies (including the CANHEART study) and sub-analyses have confirmed this association, suggesting a generalizability of these findings worldwide, in cohorts at different levels of CV risk. This aspect was tested in two independent cohorts characterized by different cardiovascular risks: the IDEAL (8888 patients with a prior coronary event randomized to receive high-intensity vs. moderate-intensity statin therapy) and EPIC-Norfolk cohorts [23] (epidemiological-based study characterized by a low risk). The U-shaped trend was confirmed in both cohorts. In the IDEAL cohort, there was an increased risk of the combined primary endpoint of coronary death, non-fatal MI, and resuscitation after cardiac arrest in patients with HDL-C levels <40 mg/dL and >80 mg/dL, whereas the lowest risk was estimated at HDL-C levels around 60–69 mg/dL. A similar pattern was observed in the EPIC cohort. Increased risk estimates were particularly present at the high end of the distributions, suggesting that these results could be driven by small subgroups of subjects with genetically determined markedly elevated HDL-C levels and characterized by dysfunctional HDL and impaired atheroprotection [24,25]. Altogether, these observations raise the question of whether a sweet spot exists for the HDL-C–CVD relationship, and whether the reasons for the increased association with CVD (and mortality) at both ends of the distribution are based on different characteristics of HDL.

## 4. Increasing Cholesterol in HDL: Pharmacological Trials

The strong and consistent association between low HDL-C levels and an increased CV risk in observational studies has prompted the search for drugs able to increase HDL-C levels; unfortunately, this effect was not mirrored by a reduction in CV risk in clinical trials evaluating such pharmacological approaches.

Niacin is one the most effective HDL-C-raising agents currently used in clinical practice. However, in the AIM-HIGH [26] and the HPS2-THRIVE trials [27], primarily designed to evaluate the effect of niacin added to a background statin on cardiovascular outcomes, the treatment with niacin increased HDL-C levels but did not lower the risk of cardiovascular events.

Fibrates are believed to increase HDL-C mainly by increasing apoA-I and apoA-II gene transcription [28]. Some data have suggested that fibrates can reduce the risk of cardiovascular events, mainly in patients who have high TG (>200 mg/dL) and low HDL-C (<35 mg/dL) levels at baseline [29]. A new fibrate, pemafibrate, a selective PPARα modulator, is currently being tested in a randomized clinical trial in patients with low baseline HDL-C and apoA-I levels, and the long-term efficacy of the drug in cardiovascular prevention is being tested in an ongoing randomized clinical trial [30].

In a meta-analysis designed to ascertain the impact of raising HDL-C on hard CV endpoints [31], including 108 randomized clinical trials (about 300,000 subjects) on statins, fibrates, niacin, and combinations of lipid-modifying drugs, no association was found between HDL-C elevation and risk of non-fatal MI, CHD mortality, or all-cause mortality after adjustment for changes in LDL-C. Of note, the mean increase in HDL-C was 1.7 mg/dL, a change that may be too small to yield significant effects on CV event rates.

Additionally, statins have been reported to affect HDL-C levels [32,33]; however, the overall effect appears to be minor, with some statins being apparently superior to others in increasing HDL-C. Nevertheless, the clinical value of this effect remains elusive.

Raising HDL also associates with the benefits of the PPAR-γ agonist pioglitazone on atherosclerosis progression in patients with type 2 diabetes, although the outcome trial, the PROactive Study, failed to show a significant reduction in the composite CV endpoint [34]. In addition, hormone replacement therapy raised plasma HDL-C but did not lower the risk of myocardial infarction [35].

The experience with CETP inhibitors has been the most challenging for the HDL hypothesis (Table 1). Torcetrapib was the first CETP inhibitor to reach advanced clinical development. In the phase III cardiovascular outcome trial (ILLUMINATE) [36], the observed increase in HDL-C was greater than 70%, with a dose-dependent effect, and accompanied by a 20% reduction in LDL-C levels. However, the development of torcetrapib was discontinued due to an increased rate of adverse events observed (including atherosclerotic cardiovascular events and total mortality). Several off-target effects of torcetrapib on blood pressure and adrenal hormone production were described, possibly explaining the adverse outcomes [37]. Three different imaging studies have also shown no benefit by torcetrapib on the progression of the carotid intima–media thickness [38,39] or coronary arteriosclerosis [40].

Another CETP inhibitor, dalcetrapib, was found to increase HDL-C levels by up to 30%, with no effect on LDL-C [41], in patients who had had an acute coronary syndrome. The cardiovascular outcome study with dalcetrapib (Dal-OUTCOMES) was discontinued for futility [42]. A post hoc pharmacogenomic analysis showed a 39% reduction in cardiovascular events and regression of atheroma at the carotid level in patients carrying the AA genotype of the ADCY9 gene on chromosome 16 [43]. This observation led to the start of a new cardiovascular outcome study with dalcetrapib in high-risk patients with the AA genotype of the ADCY9 gene [44].

Evacetrapib and anacetrapib are potent CETP inhibitors, leading to increases of up to 125% and 138%, respectively, in HDL-C, and reductions of 25–30% and 30–40%, respectively, in LDL-C levels [45,46]. However, the addition of evacetrapib to statins did not provide any additional clinical benefit, and the ACCELERATE trial was interrupted for futility [47]. In the REVEAL study [48], the addition of anacetrapib to atorvastatin reduced cardiovascular events by 9% over 4 years, but a considerable accumulation of this lipophilic drug in adipose tissue was observed. This effect, together with the modest clinical benefit, most likely driven by the beneficial effects on non-HDL-C, in line with genetic data from the general population [49,50], has hindered the approval by regulatory agencies.

A recent meta-analysis of 11 randomized controlled trials [51] examined the effects of CETP inhibitors on major cardiovascular events (MACE) and all-cause mortality, showing a non-significant reduction in the risk of non-fatal myocardial infarction (−7%) and death from cardiovascular causes (−8%).

The reasons behind these discrepancies might depend on the study design. REVEAL had a larger patient sample size, under different basal statin standards of care, and a longer follow-up vs. dal-OUTCOMES or ACCELERATE (terminated earlier for futility). This might have allowed detecting differences in CVD outcomes between treatment groups. REVEAL enrolled patients on statins at target LDL-C levels (about 60 mg/dL) at baseline, and a comparable reduction in LDL-C and apoB in the anacetrapib + atorvastatin arm as compared to atorvastatin alone was observed. Therefore, whether the CVD risk reduction might have been more likely due to non-HDL particles, thanks to the effect of statin in increasing the hepatic LDL receptor and in favoring apoB-containing lipoprotein catabolism, is a plausible conclusion [50,52]. Similar deductions come from genetics, where the combined exposure to genetic variants mimicking the action of CETP inhibitors and statins was significantly associated with a corresponding reduction in the risk of cardiovascular events that was proportional to the attenuated reduction in apoB, but significantly less than expected per unit change in LDL-C [50], perhaps due to the lack of return of cholesterol to apoB-containing lipoproteins by the inhibition of CETP.

Another possible explanation for the lack of beneficial effects of raising HDL-C based on CETP inhibition may reside in the type and characteristics of lipoproteins generated by this therapeutic approach. CETP inhibitors preferentially increase the levels of the large apoE-containing HDL particles but have little effect on the HDL particle number; CETP inhibitors act by reducing the transfer of cholesteryl esters from HDL to TG-rich lipoproteins, leading to the formation of large, cholesteryl ester-rich, mature HDL2 particles having a slower catabolism (which explains the rise in HDL-C levels), but likely less effective in exerting anti-atherogenic functions. In fact, there is evidence that cholesterol-overloaded HDL particles may exert a negative impact on the cholesterol efflux potential or a neutral, reduced hepatic selective uptake of cholesterol mediated by SR-BI, and particles are independently associated with the progression of carotid atherosclerosis [53]. The ACCENTUATE trial showed that evacetrapib significantly increased HDL-C and cholesterol efflux (both ABCA1- and non-ABCA1-mediated cholesterol efflux) but also increased apoC-III [54]; apoC-III-enriched HDL particles are less functional and might serve as a vehicle to deliver apoC-III to the arterial wall, where this protein may exert its pro-inflammatory effect. A post hoc analysis of two large prospective studies showed that very large HDL particles associated with an increased risk of cardiovascular events despite apoA-I remaining negatively associated [23]. On the other hand, apoA-I remains protective and does not become positively associated with CV risk at higher levels, which suggests that pharmacological strategies aimed at raising plasma HDL-C, but not apoA-I, levels might not be expected to have beneficial effects on atherosclerosis and, more importantly, may even increase the CV risk.

It is also worth mentioning the potential role of humoral autoimmunity as an independent CV risk factor; specifically, autoantibodies against apoA-I, which can impair the HDL function, were shown to be predictive of poor prognosis in the general population [55,56,57] as well as in high-CV risk populations [58,59,60,61]. Some studies have hypothesized that HDL-raising agents may induce a humoral response against apoA-I, thus preventing the protective activities expected by the rise in HDL-C levels. This hypothesis was confirmed in a study showing a significant and robust increase in the titers of autoantibodies against apoA-I in subjects receiving extended-release niacin [62]. In this trial, the rise in HDL-C did not parallel with an improved antioxidant capacity, likely due to the generation of autoantibodies. Several observations suggest a possible deleterious role for these autoantibodies: in vitro, anti-apoA-I autoantibodies may induce foam cell formation through a complex interplay between receptors of the innate immunity and key cholesterol homeostasis regulators, leading to the impairment of the cholesterol efflux capacity of macrophages; the presence of anti-apoA-I autoantibodies associated with an increased systemic inflammatory status in patients with myocardial infarction [63]; positive serum levels of anti-ApoA-1 autoantibodies associated with the increase in atherosclerotic plaque vulnerability in humans [64]; and, finally, anti–apoA-I autoantibodies predict major cardiovascular events in patients with rheumatoid arthritis, who are characterized by the presence of dysfunctional HDL [65]. Whether HDL-C-raising pharmacological approaches may induce the generation of anti-apoA-I autoantibodies, and whether they may exhibit a clinical relevance remain to be addressed.

## 5. HDL Cholesterol Content and Cardiovascular Events: Results from Genetic Studies

Studies of genetically determined variations in HDL-C levels [68] (without increases in plasma TGs and remnant lipoproteins) provide an ideal system in which to assess the consequences of isolated, lifelong low HDL-C levels. To determine whether genetically reduced HDL-C levels result in an increased risk of ischemic heart disease (IHD), Frikke-Schmidt et al. evaluated the association between heterozygosity for four loss-of-function mutations in ABCA1 and IHD incidence in three studies of white individuals from Copenhagen, Denmark [69]. They found that heterozygosity for loss-of-function mutations in ABCA1 was associated with a substantial, lifelong lowering of plasma levels of HDL-C (−17 mg/dL for heterozygotes vs. non-carriers, *p* < 0.001). A 17 mg/dL lower HDL-C level was associated with a multifactorially adjusted hazard ratio for IHD of 1.70 (95% confidence interval (CI), 1.57–1.85) in one of the cohorts included in the study. However, the odds ratio for IHD in heterozygotes vs. non-carriers was 0.93 (95% CI, 0.53–1.62) for the combined studies. Accordingly, functional mutations in APOA1 and LCAT associated with isolated low HDL-C did not consistently associate with an increased risk of IHD [68,70].

To further explore the possible causal relevance of HDL-C in the risk of myocardial infarction, Voight et al. performed two Mendelian randomization analyses, using both a single-nucleotide polymorphism (SNP) in the endothelial lipase gene (LIPG Asn396Ser), and a genetic score consisting of 14 common SNPs that exclusively associate with increased HDL-C [71]. Both approaches failed to show any association with a reduced risk of myocardial infarction.

Haase et al. investigated the effect of S208T (rs4986970), a variant of LCAT, in two Danish cohorts. This variant was associated with an 8 mg/dL decrease in HDL-C levels, but not with an increased risk of MI or other ischemic endpoints [72].

Taken together, these studies suggest that isolated low HDL-C is not associated with an increased risk of atherosclerotic cardiovascular events (Table 2).

The genetic modulation of CETP shows, instead, a more complex relationship with CVD. In the Copenhagen City Heart Study [76], individuals carrying four vs. no HDL-C-increasing alleles of two common genetic variants in CETP associated with reduced CETP activity showed an increase in HDL-C up to 14%, and concomitant decreases in TGs and LDL-C levels, with corresponding risk reductions in IHD (HR 0.74 (95% CI: 0.65 to 0.85)) and myocardial infarction (0.65 (95% CI: 0.54 to 0.79)). According to a Mendelian randomization study by Blauw et al. [77], higher CETP concentrations, driven by three recently identified CETP SNPs, together explaining 16.4% of the total variation in serum CETP, were associated with smaller HDL and smaller VLDL, without causal effects on LDL subclasses. In another Mendelian randomization study using genetic data on more than 100,000 participants from 14 North American and U.K. studies (1948–2012), a higher CETP score was associated with higher HDL-C, lower LDL-C, lower apoB, and a corresponding lower risk of major cardiovascular events. This reduction was similar to the reduced risk for major cardiovascular events observed with the HMGCR score, mimicking the effect of statin therapy, per unit change in LDL-C and apoB. When the CETP and HMGCR scores were combined in a factorial analysis, variants mimicking the effect of CETP inhibitors were associated with discordant reductions in LDL-C and apoB levelS, and a reduction in cardiovascular events that was proportional to the attenuated reduction in apoB but significantly less than expected per unit change in LDL-C. These results suggest that the clinical benefit of LDL-C-lowering medications may correspond more closely to apoB reductions, supporting the previous potential reason for CETP inhibitors’ failure in clinical trials [50].

## 6. Targeting HDL Function Rather Than HDL Cholesterol Content

The above-discussed evidence has led to the question of whether the cholesterol concentration in HDL might still be considered an indicator for increased risk, or whether attention should turn, instead, towards the balance, distribution, protein content, and function of distinct HDL subfractions, viewed as a more accurate indicator of the alterations in the HDL metabolic system in atherosclerosis [78].

Increased concentrations of small HDL3 and pre-β-HDL with concomitant decreased concentrations of large HDL2 are commonly observed in dyslipidemia [79]. Additionally, pre-β-HDL are significantly increased in patients with previous coronary artery disease (CAD) [70,71,72,73,74,75,76,77,78,79,80,81,82], in those with previous myocardial infarction, and in other types of patients characterized by cardiovascular co-morbidities (e.g., patients with chronic kidney disease [83]). The increase in pre-β-HDL observed in these conditions is inversely related to reduced concentrations of total α-HDL particles. This observation could be explained by either a defective activity of enzymes involved in HDL maturation [6,8,84] or an enhanced enzymatic remodeling of α-HDL induced by elevated TG levels (overproduced by the liver in patients with type IIb or atherogenic dyslipidemia). This “mass effect” of elevated cholesterol content, by altering the metabolism, size, and density of HDL [6], is of interest for the prediction of ischemic stroke [85], ischemic heart disease [86], myocardial infarction [87], and coronary artery disease [88]. Additionally, in overweight [89] and obese subjects [90], in non-obese patients with type 2 diabetes [91], and in subjects with metabolic syndrome [92], the increase in smaller HDL3 particles and the reduction in large HDL2 particles have been reported to be even more pronounced as compared to subjects with previous CAD, but without type 2 diabetes or metabolic syndrome [93] (Figure 1).

The changes in the proportion of these differently sized and dense HDL also make it difficult to draw conclusions about the relevance of the changes in the HDL apolipoprotein content during the atherosclerotic process [6]. In fact, alterations in LpA-I and LpA-I/A-II were reported in subjects with angiographically established CAD compared to control subjects [94], but no association between reduced LpA-I and the prevalence of CAD or occurrence of subsequent cardiovascular events was observed in another study [95]. Furthermore, a systemic inflammatory milieu significantly affects the proteomic content and the relative distribution within HDL subpopulations. In fact, profound changes occur in inflammatory systemic states, including alterations in the relative contents of serum amyloid A (SAA) in LpA-I:A-II in HDL isolated from uremic patients [96] and from patients with type 2 diabetes [97] (Figure 1).

Additionally, whether changes in the content of apoE and its affinity for the ABCA1/ABCG1 system are also associated with the risk of developing atherosclerotic cardiovascular diseases has been extensively debated, with contrasting results. Plasma apoE levels are inversely associated with the risk of ischemic heart disease in men, but not in women [98], and this relation is likely dependent on TG levels rather than the affinity (that is genetically determined) of apoE for its receptor. The e4 isoform of apoE, which exhibits a higher affinity for the ABCA1/ABCG1 system, was associated with a moderate increase in the risk of ischemic heart disease, while the association with the e2 isoform (characterized by a lower affinity) is uncertain, probably because of its low frequency in the population [99,100,101,102].

## 7. HDL Function/Dysfunction beyond the Reverse Cholesterol Transport

The ability of HDL particles to receive free cholesterol from cells has emerged as the culprit mechanism to explain HDL-related atheroprotective effects [103,104]; the cholesterol efflux capacity (CEC) has been shown to be inversely associated with CAD, independently of HDL-C concentrations [7,17,28,105], and with markers of preclinical atherosclerosis [29]. We must highlight the importance of taking into account all the different cholesterol-driving cellular pathways that may contribute to a specific condition. In fact, the modulation of one or more of these pathways may reflect a homeostatic feedback loop, without being a causal factor per se. This may explain some unexpected observations, such as the positive association between increased cellular cholesterol efflux and increased prospective risks for CV events [106]. This observation challenges, at least in part, the value of studying cellular cholesterol efflux assays and calls for a better understanding of what is really measured by such assays and, even more importantly, which may represent its relevance in the clinical practice. Khera et al. described a significant inverse relationship between CEC and both the carotid intima–media thickness and the risk of coronary disease, after adjustment for the HDL-C level [107]. CEC was independently and inversely associated with the presence of ASCVD [108]. In the Dallas Heart Study, CEC was associated with a significant reduction in adverse cardiovascular events (hazard ratio, 0.33; 95% CI, 0.19 to 0.55 for the highest quartile of CEC compared to the lowest quartile) after adjusting for traditional risk factors, HDL-C, and HDL particle number [17]. However, other studies failed to confirm these associations in large studies [106,109] and in peculiar populations (e.g., octogenarians [110]).

No study to date has directly demonstrated an improvement in cardiovascular outcomes by increasing CEC independently of other lipid effects [111]. As apoA-I plays a central role in cholesterol efflux, it has been suggested as a novel therapeutic target. Again, however, both Mendelian randomization studies [112] and randomized clinical trials failed to support a causal association. HDL infusion therapies transiently increase the HDL particle number and enhance the efflux capacity for cellular cholesterol. They induce a rapid, dose-proportional, and time-dependent elevation in apoA-I and pre-β-HDL particles. The first clinical trial with recombinant apoA-I Milano [113] produced a rapid regression of coronary atherosclerosis in patients following an acute coronary syndrome. However, subsequent studies failed to demonstrate incremental regression in statin-treated patients (Table 3) [114,115,116]: although this approach led to improved cholesterol efflux, its overall effect on preventing atherosclerotic disease and adverse cardiovascular events remains unclear [17,109,117].

Another strategy aiming at upregulating apoA-I has been tested with selective bromodomain and extra-terminal (BET) inhibitors, such as apabetalone, able to increase the endogenous synthesis of apoA-I by transcriptional regulation [120]. In the phase II clinical trial ASSURE, performed on patients with angiographic coronary disease and low HDL-C levels, apabetalone showed neither a greater increase in HDL-C and apoA-I levels compared to the placebo nor incremental regression of atherosclerosis while showing a higher incidence of increase in liver enzymes [121]. Recently published results of the phase III BETonMACE trial on 2425 patients with recent ACS, type 2 diabetes, and low HDL-C levels [122] showed that treatment with apabetalone added to standard therapy did not reduce the risk of cardiovascular death, non-fatal myocardial infarction, or stroke as compared with the placebo. Of note, the levels of apoA-I were a significant determinant of the cholesterol efflux capacity but accounted for less than 40% of the observed variation [107]. Given the evidence thus far, whether therapies that potently increase cholesterol efflux or alter the function of HDL particles will reduce the risk of ASCVD is still unknown.

These data highlight that alterations in HDL particles associate with cardiovascular outcomes and contribute to the refinement of the “HDL-C hypothesis”, with a greater attention to the HDL structural composition and functionality per se. This shift has been supported by multiple pieces evidence regarding the role of the apolipoprotein composition of HDL in pathophysiological mechanisms as well as in multiple acute and chronic pathological processes. For example, apoE in HDL is seminal for the extracellular lipid uptake to regulate hematopoietic cell commitment in atherosclerosis [123]. Additionally, endotoxemia alters the HDL size (decrease in the numbers of small- and medium-size particles), reduces the phospholipid and apoA-I content, and increases pro-inflammatory proteins (serum amyloid A (SAA) and secretory phospholipase A2 (sPLA2)) [124]. These structural changes in HDL particles have a higher prognostic values for sepsis than the lipoprotein cholesterol content per se [124,125]. Additionally, genetic variants in ApoL1 (encoding apolipoprotein L-1, an apolipoprotein mainly present in the HDL3 subclass, involved in ion transport, cell apoptosis, and the innate immune response against *Trypanosoma rhodesiense* [126]) predicted the onset of chronic kidney disease [127], a condition in which an impaired HDL ability in promoting cholesterol efflux, as well as alterations in the HDL apolipoprotein content, has been well documented [83,128]. In addition to renal impairment, autoimmune diseases are conditions in which alterations in the HDL system have been highlighted. In patients with systemic lupus erythematosus, for example, reduced CEC has been correlated with disease flare-up over time, carotid intima–media thickness, and cellular factors of the adaptive immune response [129,130].

It is therefore likely that the structural and dimensional balance of HDL subclasses, each perhaps endowed with some peculiar activity, rather than their cholesterol content and involvement in CEC, reflects the role of this lipoprotein class in exerting evolutionarily conserved pathophysiological functions, including atheroprotection.

## 8. Conclusions

In conclusion, the understanding of the role of HDL as a risk marker of atherosclerotic cardiovascular events is profoundly affected by both the heterogeneity of this lipoprotein class and the extensive remodeling of the HDL size and lipid and protein content during their physiological maturation. As with other acute and chronic inflammatory pathologic processes (e.g., infections, sepsis, chronic kidney disease), the complexity is even more pronounced under pathological conditions, including atherosclerosis, and in response to more complex patterns of metabolic derangement. These aspects can be accounted as the major causal factors for the lack of clear results from epidemiological data dissecting the correlation between HDL-C levels and atherosclerotic cardiovascular diseases. Genetic analyses further complicate this vision, since the analysis of mutations, or clusters of variants in loci encoding for some mediators of HDL metabolism cannot provide a clear image about the complexity of this metabolic system.

Together, uncertainties from epidemiological studies and genetics have somehow hampered an efficient pharmacological strategy, and randomized pharmacological trials targeting either HDL cholesterol or apolipoproteins, although large-sized and well conceived, did not provide convincing results. The failure of the CETP inhibitors paves the road to the rationale of the HDL mimetics, which, however, do not appear as promising as originally believed in the panorama of anti-atherosclerotic therapies.

Whether HDL is an evolutionarily conserved system accomplishing a variety of physiological functions beyond the mere cholesterol transport is a plausible possibility to be taken into consideration back to the bench and to be harnessed for tailoring alternative pharmacological options.

## Figures and Tables

**Figure 1 cells-10-01869-f001:**
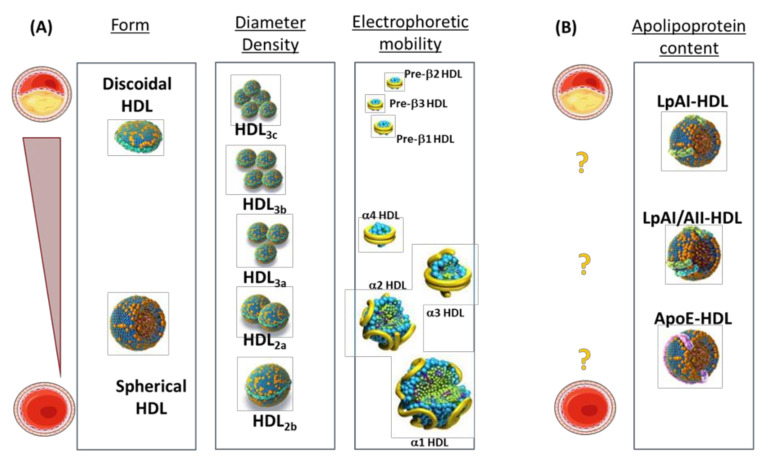
Heterogeneity of the HDL class in atherosclerotic cardiovascular diseases. (**A**) HDL major types by form, diameters/density, or electrophoretic mobility, and their known association with risk of atherosclerotic cardiovascular disease (reporting increase in the risk from bottom to top). (**B**) Summary of unknown/uncertain associations between apolipoprotein content of HDL and risk of atherosclerotic cardiovascular disease (indicated by yellow question marks).

**Table 1 cells-10-01869-t001:** CETP inhibitors: clinical trials and cardiovascular outcomes.

Ref.	CETP Inhibitor	Trial	Patients	Duration	Effect on Lipids in Treatment Group	CV Results
Barter PJ. N. Engl. J. Med. 2007 [66]	Torcetrapib	ILLUMINATE	15,067 patients at high cardiovascular risk	1–2 years	HDL-C: +72% LDL-C: −25%	Increased risk of cardiovascular events (HR 1.25; 95%CI 1.09–1.44; *p =* 0.001) and death from any cause (HR 1.58; 95%CI 1.14–2.19; *p =* 0.006)
Schwartz GG. N. Engl. J. Med. 2012 [42]	Dalcetrapib	Dal-OUTCOMES	15,871 patients recently hospitalized for acute coronary syndrome	31 months (stopped early for futility)	HDL-C: 31 to 40% minimal effect on LDL-C	No effect on the CV composite endpoint (HR 1.04; 95%CI 0.93–1.16; *p =* 0.52)
Lincoff AM. N. Engl. J. Med. 2017 [47]	Evacetrapib	ACCELERATE	12,092 patients at high cardiovascular risk	26 months (stopped early for futility)	HDL-C: +133% LDL-C: −31%	No effect on the CV composite endpoint (HR 1.01; 95%CI 0.91–1.11; *p =* 0.91)
Bowman L. N. Engl. J. Med. 2017 [67]	Anacetrapib	REVEAL	30,449 patients at high cardiovascular risk	4.1 years	HDL-C: +104% LDL-C: −26%	Decrease in the CV composite endpoint (HR 0.91; 95%CI 0.85–0.97; *p =* 0.004)

HR: hazard ratio.

**Table 2 cells-10-01869-t002:** Mendelian randomization studies about HDL and cardiovascular events.

Publication	Gene	Effect on HDL-C Levels	CV Outcome	Expected Effect on CV Outcome	Observed Effect on CV Outcome
Frikke-Schmidt R. 2008 [69]	*ABCA1*	−17 mg/dL	ischemic heart disease	HR 1.70 (95%CI 1.57–1.85)	OR 0.93 (95%CI 0.53–1.62)
Johannsen TH. 2009 [73]	*LIPC*	+16% (8 mg/dL)	ischemic heart disease	HR 0.87 (95%CI 0.84–0.90)	OR 1.19 (95%CI 0.76–1.88)
Haase CL. 2010 [74]	*APOA1*	+7.7%	ischemic heart disease myocardial infarction	HR 0.93 (95%CI 0.91–0.94) HR 0.89 (95%CI 0.86–0.92)	OR 1.10 (95%CI 0.89–1.35) OR 1.14 (95%CI 0.89–1.46)
Voight BF. 2012 [71]	*LIPG*	+5.5 mg/dL	myocardial infarction	OR 0.87 (95%CI 0.84–0.91)	OR 0.99 (95%CI 0.88–1.11)
Haase CL. 2012 [72]	*LCAT*	−13% (8 mg/dL)	myocardial infarction	HR 1.18 (95%CI 1.12–1.24)	OR 0.53 (95%CI 0.92–1.25)
Voight BF. 2012 [71]	14 SNP score	-	myocardial infarction	OR 0.62 (95%CI 0.58–0.66) *	OR 0.93 (95%CI 0.68–1.26) *
Holmes MV. 2015 [75]	19 SNP score	8.9 mg/dL ^	myocardial infarction	-	OR 0.91 (95%CI 0.42–1.98) **
	48 SNP score	3.1 mg/dL ^	myocardial infarction	-	OR 0.81 (95%CI 0.44–1.46) **

^ mean difference comparing top to bottom quintiles of each allele score; * per 1 SD increase in HDL-C due to SNP score; ** per 1 mmol/L higher HDL-C. OR: odds ratio; HR hazard ratio. Genes. ABCA1: ATP-binding cassette transporter A1; APOA1: apolipoprotein A-I; LCAT: lecithin-cholesterol acyl transferase; LIPC: hepatic lipase; LIPG: endothelial lipase; SCARB1: scavenger receptor class B type 1.

**Table 3 cells-10-01869-t003:** HDL mimetics: clinical trials and cardiovascular outcomes.

Ref.	HDL Mimetic	Apo A1	Trial	Patients	Effect on Cholesterol Efflux	CV Outcome	Results
Michael Gibson C. 2016 [118]	CSL-112	wild-type apoA-I	AEGIS-I trial	1258 patients with a recent acute myocardial infarction	increased apoA-I and ex vivo cholesterol efflux	time to first occurrence of a MACE **	HR 1.02 (95%CI, 0.57–1.80; *p =* 0.52)
Tardif J-C. 2014 [119]	CER-001	apoA-I and sphingomyelin	CHI-SQUARE study	507 patients with a clinical indication for coronary angiography	increased cholesterol mobilization	PAV *	0.02% with placebo and 0.19%; with CER-001 (difference, *p* = 0.53)
Nicholls SJ. 2018 [116]	MDCO-216	apoA-I Milano	MILANO-PILOT Trial	122 post-ACS patients on optimal conventional medical treatment	increased ATP-binding cassette transporter A1-mediated cholesterol efflux	PAV *	−0.94% with placebo and −0.21% with MDCO-216 (difference, 0.73%; 95%CI, −0.07 to 1.52; *p* = 0.07)
Nicholls SJ. JAMA Cardiol. 2018 [115]	CER-001	apoA-I and sphingomyelin	CARAT study	293 patients with status post-ACS	increased cholesterol mobilization	PAV*	−0.41% with placebo and −0.09%; with CER-001 (difference 0.32%; *p =* 0.15)

* primary outcome; ** secondary outcome. HR hazard ratio; PAV: percent atheroma volume; MACE: major adverse cardiac events.

## Data Availability

Not applicable.

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
