# Peer review of "HDL in Atherosclerotic Cardiovascular Disease: In Search of a Role"

_cells, 2021, doi:10.3390/cells10081869_

Round 1

Reviewer 1 Report

The manuscript by Casula and colleagues provides a timely and comprehensive introduction to the current understanding of anti-atherogenic function of HDL and its potential therapeutic applications. As far as I can judge, the manuscript covers most recent advances in the field including interventional studies currently in progress. It is very well-written and easy to follow and understandable also for non-specialists in the field. It also contains helpful tables and is well-illustrated. My only criticism is that the concept of dysfunctional HDL, which is crucial to the proper explanation, why pharmacological interventions based on elevation of HDL-cholesterol were likely on the wrong track, is presented in a slightly cursory fashion and does not get its fair share. I would like to suggest, therefore, that the penultimate section devoted to this concept is significantly expanded and perhaps enriched with a figure or table.

Author Response

Dear reviewer#1,

thank you very much for your positive evaluation of our review and for the constrictive comment, which helped us to add relevant aspect for the discussion of this topic.

Please, find below in red the answers to the point you raised.

The manuscript by Casula and colleagues provides a timely and comprehensive introduction to the current understanding of anti-atherogenic function of HDL and its potential therapeutic applications. As far as I can judge, the manuscript covers most recent advances in the field including interventional studies currently in progress. It is very well-written and easy to follow and understandable also for non-specialists in the field. It also contains helpful tables and is well-illustrated. My only criticism is that the concept of dysfunctional HDL, which is crucial to the proper explanation, why pharmacological interventions based on elevation of HDL-cholesterol were likely on the wrong track, is presented in a slightly cursory fashion and does not get its fair share. I would like to suggest, therefore, that the penultimate section devoted to this concept is significantly expanded and perhaps enriched with a figure or table.

We thank the reviewer for her/his positive evaluation of the general manuscript.

According to reviewer’s indication, we now included in the penultimate section (page 11, lines 417-425) this following sentence:

“… We must highlight the importance of taking into account all the different cholesterol driving cellular pathways that may contribute in a specific condition. In fact, the modulation of one or more of these pathways may reflect a homeostatic feedback loop, without being per se a causal factor. This may explain some unexpected observations, such as the positive association between increased cellular cholesterol efflux and in-creased prospective risks for CV events [106]. This observation challenges, at least in part, the value of studying cellular cholesterol efflux assays and calls for a better understanding of what is really measured by such assays and, even more importantly, which may be its relevance in the clinical practice…”

References:

  1. Li, X.M.; Tang, W.H.; Mosior, M.K.; Huang, Y.; Wu, Y.; Matter, W.; Gao, V.; Schmitt, D.; Didonato, J.A.; Fisher, E.A.; et al. Paradoxical association of enhanced cho-lesterol efflux with increased incident cardiovascular risks. Arteriosclerosis, throm-bosis, and vascular biology 2013, 33, 1696-1705, doi:10.1161/ATVBAHA.113.301373.

Reviewer 2 Report

This well-written review certainly addresses an important topic worth to be specifically reported in the Journal. There are however several points of attention needing to be addressed and improved before considering this manuscript as suitable for publication.

Major Comments:

Two important missing elements that could increase the novelty of the present review in this already crowded field have to be considered.

  1. Firstly, most to the studies performed so far focused either on ABCA1, ABCG1-mediated, or total CEC without being systematically performed in parallel, and without taking passive diffusion into account. Furthermore no precision is provided nor discussed regarding the kind of model the CEC is performed (macropahges if so which ones, fibroblasts, others?). As these are currently a highly unstandardized measures, these details matter a lot. Furthremore, if ABCA1, ABCG1 pathways are becoming predominant in highly dyslipidemic situations, they are not in normocholesterolemic settings where passive diffusion is the key driver of efflux (Adorni MP et al.;  Lipid Res.2007, 48, 2453–2462; Phillips MC. J. Biolog. Chem. 2014, 289, 24020–24029; Phillips MC et al. Atherosclerosis 1998, 137, S13–S17). This highlights the fact that not only it is required to specify the kind of CEC the authors are referring to throughout the manuscript (by far not the case presently), but also emphasize the importance of taking into account all the different cholesterol driving cellular fluxes at play in a given situation in order to apprehend correctly the overall picture. An activation of a given pathway could just the reflect of a homeostatic feedback loops without being per se the causal factor, indicating that the risk of “reverse causality” interpretations may well be substantial in the field and underlie some of the contradictory results regarding ABCA1-efflux and CVD risk (Li XM et al. Arterioscler. Thromb. Vasc. Biol. 2013, 33, 1696–1705). These points need to be further presented and developped  in the review. 

Secondly and in possibly relation with the aforementioned point as well as as with the announced scope of the review, some HDL-raising therapeutics (niacin in particular) have been shown to increase the humoral response against ApoA-1, mediating a loss of HDL anti-oxidant function according to the randomized phase II Explore trial (Batuca JR et al.; Br. J. Clin. Pharmacol. 2017, 83, 1002–1010). Furthermore, as these autoantibodies have been shown to disrupt cellular macrophages cholesterol homeostasis leading to foam cell formation by increasing LDL-R, HMCOA reductase expression and ACAT-1 activity despite increasing ABCA-1 efflux (Pagano S et al. J Clin Med. 2019;8(12):2035.Vuilleumier N et al. J Clin Med. 2019;8(8):1225, the relevance of those autoantibodies in field of HDL function needs to be addressed, especially as several of the HDL functionalities have been shown to be impaired in autoimmune diseases where anti-apoA-1 IgG are raised (see for example Ames PR, et al. Lupus. 2010;19(6):711-6. Nigolian H, et al. Rheumatology (Oxford) 2020;59(3):534-544; Vuilleumier N,  et al. Arthritis Rheum. 2010;62(9):2640-50. Chistiakov DA et al.Lab Invest. 2016;96(7):708-18) and potentially causally related to a loss of HDL anti-oxidant properties (Batuca JR et al.; Br. J. Clin. Pharmacol. 2017, 83, 1002–1010; Ames PR et al. Lupus 2010;19(6):711-16). Regardless of whether this pharmacologically-induced autoimmune phenomenon is a generic phenomenon ascribed to all HDL-raising therapies or whether it’s specific to niacin which would be worth further explorations, the data related to anti-apoA-1 IgG and HDL function needs to be introduced in the review.

Minor comments:

-First line of abstract: “vs” to be replaced by « regarding the » or a similar wording.

-Second line of abstract: CVD) must be (CVD)

Author Response

Dear reviewer#2,

thank you very much for your positive and constructive comments, which helped us to add relevant aspect for the discussion of this topic.

Please, find below in red the answers to both the major points that were raised.

Firstly, most to the studies performed so far focused either on ABCA1, ABCG1-mediated, or total CEC without being systematically performed in parallel, and without taking passive diffusion into account. Furthermore no precision is provided nor discussed regarding the kind of model the CEC is performed (macropahges if so which ones, fibroblasts, others?). As these are currently a highly unstandardized measures, these details matter a lot. Furthremore, if ABCA1, ABCG1 pathways are becoming predominant in highly dyslipidemic situations, they are not in normocholesterolemic settings where passive diffusion is the key driver of efflux (Adorni MP et al.;  Lipid Res.2007, 48, 2453–2462; Phillips MC. J. Biolog. Chem. 2014, 289, 24020–24029; Phillips MC et al. Atherosclerosis 1998, 137, S13–S17). This highlights the fact that not only it is required to specify the kind of CEC the authors are referring to throughout the manuscript (by far not the case presently), but also emphasize the importance of taking into account all the different cholesterol driving cellular fluxes at play in a given situation in order to apprehend correctly the overall picture. An activation of a given pathway could just the reflect of a homeostatic feedback loops without being per se the causal factor, indicating that the risk of “reverse causality” interpretations may well be substantial in the field and underlie some of the contradictory results regarding ABCA1-efflux and CVD risk (Li XM et al. Arterioscler. Thromb. Vasc. Biol. 2013, 33, 1696–1705). These points need to be further presented and developped  in the review.

We agree with the reviewer and we now included this sentence (pages 5-6, lines 244-265), as follows:

“...Another possible explanation for the lack of beneficial effect of HDL-C-raising based on CETP inhibition may reside in the type and characteristics of lipoproteins generated by this therapeutic approach. CETP inhibitors preferentially increase the levels of the large apoE-containing HDL particles, but have little effect on HDL particle number; CETP inhibitors act by reducing the transfer of cholesteryl esters from HDL to TG-rich lipoproteins, leading to the formation of large, cholesteryl ester-rich, mature HDL2 particles having a slower catabolism (which explain the rise in HDL-C levels), but likely less effective in exerting anti-atherogenic functions. In fact, there is evidence that cholesterol-overloaded HDL particles may exert a negative impact on the cholesterol efflux potential or are neutral, reduced hepatic selective uptake of cholesterol mediated by SR-BI, and were particles are independently associated with the progression of ca-rotid atherosclerosis [53]. The ACCENTUATE trial showed that evacetrapib increased significantly HDL-C and cholesterol efflux (both ABCA1- and non-ABCA1-mediated cholesterol efflux), but also increased apoC-III [54]; apoC-III-enriched HDL particles are less functional and might serve as a vehicle to deliver apoC-III to the arterial wall, where this protein may exert its pro-inflammatory effect. A post-hoc analysis of 2 large prospective studies showed that very large HDL particles associated with an increased risk of cardiovascular events despite apoA-I remained negatively associated [23]. On the other hand, apoA-I remains protective and does not become positively associated with CV risk at higher levels, which suggests that pharmacological strategies aimed at raising plasma HDL-C but not apoA-I levels might not be expected to have beneficial effects on atherosclerosis and, more importantly, may even increase the CV risk….”.

References:

  1. van der Steeg, W.A.; Holme, I.; Boekholdt, S.M.; Larsen, M.L.; Lindahl, C.; Stroes, E.S.; Tikkanen, M.J.; Wareham, N.J.; Faergeman, O.; Olsson, A.G.; et al. High-density lipoprotein cholesterol, high-density lipoprotein particle size, and apolipoprotein A-I: significance for cardiovascular risk: the IDEAL and EPIC-Norfolk studies. Journal of the American College of Cardiology 2008, 51, 634-642, doi:10.1016/j.jacc.2007.09.060.
  2. Qi, Y.; Fan, J.; Liu, J.; Wang, W.; Wang, M.; Sun, J.; Liu, J.; Xie, W.; Zhao, F.; Li, Y.; et al. Cholesterol-overloaded HDL particles are independently associated with progres-sion of carotid atherosclerosis in a cardiovascular disease-free population: a communi-ty-based cohort study. J Am Coll Cardiol 2015, 65, 355-363, doi:10.1016/j.jacc.2014.11.019
  3. Nicholls, S.J.; Ray, K.K.; Ballantyne, C.M.; Beacham, L.A.; Miller, D.L.; Ruotolo, G.; Nissen, S.E.; Riesmeyer, J.S.; Investigators, A. Comparative effects of cholesteryl ester transfer protein inhibition, statin or ezetimibe on lipid factors: The ACCENTUATE tri-al. Atherosclerosis 2017, 261, 12-18, doi:10.1016/j.atherosclerosis.2017.04.008.

Secondly and in possibly relation with the aforementioned point as well as as with the announced scope of the review, some HDL-raising therapeutics (niacin in particular) have been shown to increase the humoral response against ApoA-1, mediating a loss of HDL anti-oxidant function according to the randomized phase II Explore trial (Batuca JR et al.; Br. J. Clin. Pharmacol. 2017, 83, 1002–1010). Furthermore, as these autoantibodies have been shown to disrupt cellular macrophages cholesterol homeostasis leading to foam cell formation by increasing LDL-R, HMCOA reductase expression and ACAT-1 activity despite increasing ABCA-1 efflux (Pagano S et al. J Clin Med. 2019;8(12):2035.Vuilleumier N et al. J Clin Med. 2019;8(8):1225, the relevance of those autoantibodies in field of HDL function needs to be addressed, especially as several of the HDL functionalities have been shown to be impaired in autoimmune diseases where anti-apoA-1 IgG are raised (see for example Ames PR, et al. Lupus. 2010;19(6):711-6. Nigolian H, et al. Rheumatology (Oxford) 2020;59(3):534-544; Vuilleumier N,  et al. Arthritis Rheum. 2010;62(9):2640-50. Chistiakov DA et al.Lab Invest. 2016;96(7):708-18) and potentially causally related to a loss of HDL anti-oxidant properties (Batuca JR et al.; Br. J. Clin. Pharmacol. 2017, 83, 1002–1010; Ames PR et al. Lupus 2010;19(6):711-16). Regardless of whether this pharmacologically-induced autoimmune phenomenon is a generic phenomenon ascribed to all HDL-raising therapies or whether it’s specific to niacin which would be worth further explorations, the data related to anti-apoA-1 IgG and HDL function needs to be introduced in the review.

Following this valuable suggestion, we included this sentence (page 6, lines 266-286):

“…It is worth mentioning also the potential role of humoral autoimmunity as an independent CV risk factor; specifically, autoantibodies against apoA-I, that can impair HDL function, were shown to be predictive of poor prognosis in the general population [55-57] as well as in high CV risk populations [58-61]. Some studies have hypothesized that HDL-raising agents may induce a humoral response against apoA-I, thus preventing the protective activities expected by the rise in HDL-C levels. This hypothesis was confirmed in a study showing a significant and robust increase of the titres of autoantibodies against apoA-I in subjects receiving extended-release niacin [62]. In this trial, the rise in HDL-C did not parallel with an improved antioxidant capacity, likely due to the generation of autoantibodies. Several observations suggest a possible deleterious role for these autoantibodies: in vitro, anti-apoA-I autoantibodies may induce foam cell formation through a complex interplay between receptors of innate immunity and key cholesterol homeostasis regulators, leading to the impairment of the cholesterol efflux capacity of macrophages; the presence of anti-apoA-I autoantibodies associated with an increased systemic inflammatory status in patients with myocardial infarction [63]; positive serum levels of anti-ApoA-1 autoantibodies associated with the increase of atherosclerotic plaque vulnerability in humans [64]; finally, anti–apoA-I autoantibodies predict major cardiovascular events in patients with rheumatoid arthritis, who are characterized by the presence of dysfunctional HDL [65]. Whether HDL-C-raising pharmacological approaches may induce the generation of anti-apoA-I autoantibodies and whether they may exhibit a clinical relevance remains to be ad-dressed…”.

References:

  1. Antiochos, P.; Marques-Vidal, P.; Virzi, J.; Pagano, S.; Satta, N.; Hartley, O.; Mon-tecucco, F.; Mach, F.; Kutalik, Z.; Waeber, G.; et al. Impact of CD14 Polymorphisms on Anti-Apolipoprotein A-1 IgG-Related Coronary Artery Disease Prediction in the Gen-eral Population. Arterioscler Thromb Vasc Biol 2017, 37, 2342-2349, doi:10.1161/ATVBAHA.117.309602.
  2. Antiochos, P.; Marques-Vidal, P.; Virzi, J.; Pagano, S.; Satta, N.; Bastardot, F.; Hartley, O.; Montecucco, F.; Mach, F.; Waeber, G.; et al. Association between an-ti-apolipoprotein A-1 antibodies and cardiovascular disease in the general population. Results from the CoLaus study. Thromb Haemost 2016, 116, 764-771, doi:10.1160/TH16-03-0248.
  3. Antiochos, P.; Marques-Vidal, P.; Virzi, J.; Pagano, S.; Satta, N.; Hartley, O.; Mon-tecucco, F.; Mach, F.; Kutalik, Z.; Waeber, G.; et al. Anti-Apolipoprotein A-1 IgG Predict All-Cause Mortality and Are Associated with Fc Receptor-Like 3 Polymorphisms. Front Immunol 2017, 8, 437, doi:10.3389/fimmu.2017.00437.
  4. Carbone, F.; Satta, N.; Montecucco, F.; Virzi, J.; Burger, F.; Roth, A.; Roversi, G.; Tamborino, C.; Casetta, I.; Seraceni, S.; et al. Anti-ApoA-1 IgG serum levels predict worse poststroke outcomes. Eur J Clin Invest 2016, 46, 805-817, doi:10.1111/eci.12664.
  5. Vuilleumier, N.; Montecucco, F.; Spinella, G.; Pagano, S.; Bertolotto, M.; Pane, B.; Pende, A.; Galan, K.; Roux-Lombard, P.; Combescure, C.; et al. Serum levels of an-ti-apolipoprotein A-1 auto-antibodies and myeloperoxidase as predictors of major ad-verse cardiovascular events after carotid endarterectomy. Thromb Haemost 2013, 109, 706-715, doi:10.1160/TH12-10-0714.
  6. Anderson, J.L.C.; Pagano, S.; Virzi, J.; Dullaart, R.P.F.; Annema, W.; Kuipers, F.; Bakker, S.J.L.; Vuilleumier, N.; Tietge, U.J.F. Autoantibodies to Apolipoprotein A-1 as Independent Predictors of Cardiovascular Mortality in Renal Transplant Recipients. J Clin Med 2019, 8, doi:10.3390/jcm8070948.
  7. Vuilleumier, N.; Pagano, S.; Combescure, C.; Gencer, B.; Virzi, J.; Raber, L.; Car-ballo, D.; Carballo, S.; Nanchen, D.; Rodondi, N.; et al. Non-Linear Relationship be-tween Anti-Apolipoprotein A-1 IgGs and Cardiovascular Outcomes in Patients with Acute Coronary Syndromes. J Clin Med 2019, 8, doi:10.3390/jcm8071002.
  8. Batuca, J.R.; Amaral, M.C.; Favas, C.; Paula, F.S.; Ames, P.R.J.; Papoila, A.L.; Del-gado Alves, J. Extended-release niacin increases anti-apolipoprotein A-I antibodies that block the antioxidant effect of high-density lipoprotein-cholesterol: the EXPLORE clinical trial. Br J Clin Pharmacol 2017, 83, 1002-1010, doi:10.1111/bcp.13198.
  9. Pagano, S.; Satta, N.; Werling, D.; Offord, V.; de Moerloose, P.; Charbonney, E.; Hochstrasser, D.; Roux-Lombard, P.; Vuilleumier, N. Anti-apolipoprotein A-1 IgG in patients with myocardial infarction promotes inflammation through TLR2/CD14 com-plex. J Intern Med 2012, 272, 344-357, doi:10.1111/j.1365-2796.2012.02530.x.
  10. Montecucco, F.; Vuilleumier, N.; Pagano, S.; Lenglet, S.; Bertolotto, M.; Brau-nersreuther, V.; Pelli, G.; Kovari, E.; Pane, B.; Spinella, G.; et al. Anti-Apolipoprotein A-1 auto-antibodies are active mediators of atherosclerotic plaque vulnerability. Eur Heart J 2011, 32, 412-421, doi:10.1093/eurheartj/ehq521.
  11. Vuilleumier, N.; Bas, S.; Pagano, S.; Montecucco, F.; Guerne, P.A.; Finckh, A.; Lovis, C.; Mach, F.; Hochstrasser, D.; Roux-Lombard, P.; et al. Anti-apolipoprotein A-1 IgG predicts major cardiovascular events in patients with rheumatoid arthritis. Arthri-tis Rheum 2010, 62, 2640-2650, doi:10.1002/art.27546.

Minor comments:

-First line of abstract: “vs” to be replaced by « regarding the » or a similar wording.

This sentence has been revised in the abstract, which has been revised as follows:

“..Abstract: For long time high-density lipoprotein cholesterol (HDL-C) has been regarded as a cardiovascular disease (CVD) protective factor. Recently several epidemiological studies, while confirming low plasma levels of HDL-C as an established predictive biomarker for atherosclerotic CVD, indicate that not only people at the lowest, but also those with high HDL-C are at increased risk of cardiovascular (CV) mortality…”.

-Second line of abstract: CVD) must be (CVD)

We now checked for this typo in the abstract. We additionally checked the entire text of the manuscript for grammar errors.

Round 2

Reviewer 2 Report

All my queries have been adressed